# Retinal Disease Variability in Female Carriers of *RPGR* Variants Associated with Retinitis Pigmentosa: Clinical and Genetic Parameters

**DOI:** 10.3390/genes16020221

**Published:** 2025-02-13

**Authors:** Sena A. Gocuk, Thomas L. Edwards, Jasleen K. Jolly, Fred K. Chen, David C. Sousa, Myra B. McGuinness, Terri L. McLaren, Tina M. Lamey, Jennifer A. Thompson, Lauren N. Ayton

**Affiliations:** 1Department of Optometry and Vision Sciences, University of Melbourne, Melbourne, VIC 3001, Australia; layton@unimelb.edu.au; 2Ophthalmology, Department of Surgery, University of Melbourne, Melbourne, VIC 3001, Australia; tom.edwards@unimelb.edu.au (T.L.E.); fred.chen@unimelb.edu.au (F.K.C.); davidscsousa@gmail.com (D.C.S.); myra.mcguinness@unimelb.edu.au (M.B.M.); 3Centre for Eye Research Australia, Royal Victorian Eye and Ear Hospital, Melbourne, VIC 3002, Australia; 4Vision and Eye Research Institute, Anglia Ruskin University, Cambridge CB1 1PT, UK; jasleen.jolly@unimelb.edu.au; 5Centre for Ophthalmology and Visual Science (Incorporating Lions Eye Institute), The University of Western Australia, Nedlands, WA 6009, Australia; terri.mclaren@health.wa.gov.au; 6Australian Inherited Retinal Disease Registry and DNA Bank, Department of Medical Technology and Physics, Sir Charles Gairdner Hospital, Hospital Avenue, Nedlands, WA 6009, Australia; tina.lamey@health.wa.gov.au (T.M.L.); jennifer.thompson3@health.wa.gov.au (J.A.T.); 7Centre for Epidemiology and Biostatistics, Melbourne School of Population and Global Health, University of Melbourne, Melbourne, VIC 3001, Australia

**Keywords:** OCT, cross-sectional, correlation, ORF15, genotype, LLVA

## Abstract

*Objectives:* We sought to investigate the visual function, retinal features, and genotype–phenotype correlations of an Australian cohort of *RPGR* carriers. *Methods:* In this cross-sectional study, we evaluated *RPGR* carriers seen in Melbourne and Perth between 2013 and 2023 and healthy women seen between 2022 and 2023 in Melbourne. Visual acuity tests, fundus-tracked microperimetry, and retinal imaging were performed. *RPGR* carriers were classified into four retinal phenotypes (normal, radial, focal pigmentary retinopathy, and male pattern phenotype) and compared against healthy controls. Genotype–phenotype relationships in the RPGR carriers were investigated. *Results:* Thirty-five female *RPGR* carriers and thirty healthy controls were included in this study. The median ages were 40 and 48.5 years for *RPGR* carriers and controls, respectively (*p* = 0.26). Most *RPGR* carriers (89%) had a genetic diagnosis. Best-corrected visual acuity (BCVA), low luminance visual acuity, retinal sensitivity, central inner retinal thickness (IRT, 1°), and photoreceptor complex (PRC) thickness across the central 1–7° of the retina differed between phenotypes of *RPGR* carriers. On average, *RPGR* carriers with ORF15 variants (n = 25 carriers) had reduced LLVA, a greater IRT at 1°, and thinner PRC thickness at 7° from the fovea (all *p* < 0.05) compared to those with exon 1–14 variants. *Conclusions:* Female *RPGR* carriers with severe retinal phenotypes had significantly decreased visual function and changes in retinal structure in comparison to both the controls and carriers with mild retinal disease. BCVA, LLVA, retinal sensitivity, and retinal thickness are biomarkers for detecting retinal disease in *RPGR* carriers. The genetic variant alone did not influence retinal phenotype; however, *RPGR* carriers with ORF15 variants exhibited reduced retinal and visual measurements compared to those with exon 1–14 variants.

## 1. Introduction

Retinitis pigmentosa (RP), or rod–cone dystrophy, is a heterogeneous group of disorders typically caused by primary degeneration of the rod photoreceptors followed by the cones in later stages of the condition [1]. Among the various inheritance patterns, X-linked RP (XLRP) is the least common; however, it leads to the most severe presentation in affected individuals [2]. In early childhood, affected males experience high myopia, night blindness (nyctalopia), and peripheral vision deterioration, which progresses to central vision loss in their third decade [2].

There are currently over 60 genes associated with RP, and at least four are associated with XLRP: *RPGR*, *RP2*, and *OFD1*. Mutations in the *RPGR* gene account for over 70% of presentations of XLRP [3]. *RPGR*-associated retinal dystrophy presents a heterogeneous phenotype in males, ranging from panretinal rod–cone dystrophy to predominant cone dystrophy [1,3].

Female carriers of X-linked RP associated with *RPGR* variants (referred to as *RPGR* carriers hereafter) exhibit variable retinal disease severity, ranging from subretinal normal retinae to severe degeneration; the latter is known as the “male pattern” phenotype [4]. In 2018, Nanda et al. classified retinal disease severity in *RPGR* carriers, based on fundus autofluorescence (FAF) imaging, as normal, radial pattern, focal pigmentary retinopathy, and male pattern phenotypes [4]. This variability is due to X-chromosome inactivation, i.e., the random silencing of one of the two X chromosomes, which occurs during early embryonic development [5,6].

Previous studies have utilised small samples to examine the retinal characteristics of *RPGR* carriers, resulting in conflicting evidence regarding disease biomarkers. In particular, there has been discordance in reports of the differences in retinal thickness and outer-retinal alterations between *RPGR* carriers and healthy controls [7,8,9]. Previous studies also rarely report the severity of retinal disease among study participants, a crucial factor for accurate interpretation, given the wide spectrum observed.

There is now substantial evidence supporting retinal remodelling in degenerative retinal diseases, a process involving glial and vascular reorganisation, triggered by the loss of photoreceptors [10]. These alterations may commence as early as the onset of photoreceptor stress in the retina, raising a key question regarding the optimal timing for therapeutic intervention. Jolly et al. investigated retinal remodelling in males with *RPGR*-associated RP by showing inner-retinal thickening concurrent with photoreceptor thinning [11]. Consequently, the researchers proposed that assessing the thickness of the inner retina and photoreceptor complex might offer greater sensitivity in detecting retinal changes compared to total retinal thickness measurements. However, this has not been investigated in relation to *RPGR* carriers of varying retinal disease severity.

Insights into genetic variants and their phenotypic expression have come from the genetic testing of people with RP. Disease-causing variants have been identified across exons 1–15 in the *RPGR* gene [12]. Up to 73% of mutations occur in the ORF15 region, which is regarded as the mutational hotspot of the *RPGR* gene due to a glutamate–glycine-rich domain at this site, thus leading to genomic instability [1,13]. As our understanding of the *RPGR* gene has expanded, there has been considerable interest in elucidating genotype–phenotype correlations for this condition. There are conflicting reports of genotype–phenotype associations relating to genetic variant location and disease severity in affected males and females [2,14,15,16].

Recent developments in retinal gene therapy have sparked interest in learning more about *RPGR* variants. Currently, there are eight gene therapy clinical trials for *RPGR*-associated XLRP registered on clinicaltrials.gov; however, females are only eligible to enrol in two of these studies. Furthermore, there are no details pertaining to which carriers and degrees of retinal disease are eligible. As such, it is crucial to identify sensitive retinal biomarkers for detecting and monitoring retinal disease and accurately assess the suitability of *RPGR* carriers for upcoming therapeutic interventions.

This cross-sectional study was based in two Australian sites (Melbourne and Perth) and assessed a comparatively large cohort of *RPGR* carriers and healthy controls to describe the retinal disease spectrum using retinal imaging and microperimetry. The sim of this study was to investigate the association between genetic variants (i.e., the type and location of variants in the *RPGR* gene) and subsequent retinal phenotypes and/or retinal and visual biomarkers. This information will guide the clinical management of *RPGR* carriers and help reveal the most appropriate clinical tests for detecting milder disease.

## 2. Methods

The *RPGR* carriers in this cross-sectional study were recruited prospectively, using the same methods and equipment at both sites. Healthy controls were recruited as part of a large natural history study at the primary site (Melbourne, Australia) between 2022 and 2023. The research methods employed and healthy control data were previously published in a study investigating female carriers of choroideremia (*CHM*) [17].

All participants were recruited via advertisements distributed by clinicians, social media, support groups, and patient associations (see acknowledgments). Research visits were conducted between 2013 and 2023 (Perth) and 2021 and 2023 (Melbourne). All participants gave written informed consent before their research visits. This study adheres to the Declaration of Helsinki and complies with institutional ethics and governance guidelines (Melbourne: The Royal Victorian Eye and Ear Hospital Human Research Ethics Committee: 19-1443H, Perth: The University of Western Australia Human Ethics Office of Research: 2021/ET000151).

### 2.1. Eligibility

This study included female *RPGR* carriers, aged 18 years or older, confirmed through genetic testing or identified as obligate carriers (based on definitive family history of *RPGR*-associated XLRP). Age of diagnosis, severity of retinal disease, or presence of other ocular conditions did not limit inclusion in this study, except for conditions known to affect macular health. Healthy controls aged 18 years or older with no known ocular conditions were also invited to participate in this study. While genetic testing was not conducted to confirm the absence of inherited retinal diseases (IRDs), in the selection of healthy controls, individuals with a known family history were excluded.

### 2.2. Clinical Tests

Clinical tests performed in this study formed part of a large natural history study focused on female carriers of X-linked IRDs, the protocol for which was previously published [17]. All participants were subjected to detailed clinical history taking, subjective refraction, anterior eye examinations, fundus-tracked microperimetry, and retinal imaging (these procedures are summarised in Appendix A).

Retinal disease was classified into four retinal disease severities based on FAF imaging, as defined by Nanda et al. [4]: normal (N), radial (R) pattern reflex without pigmentary retinopathy, focal (F) pigmentary retinopathy, and male (M) phenotype (Figure 1). This classification represents the state of disease at the time of the research visit, acknowledging that carriers can be at different stages of disease progression. Since this is a cross-sectional study, it is crucial to interpret the retinal phenotype in the context of the participant’s age and disease development stage. Two independent graders (SAG, an optometry-trained researcher with expertise in RPGR-associated XLRP, and DCS, an ophthalmology-trained retinal specialist) classified retinal disease severity based on 55-degree FAF imaging. Any disagreement in regard to retinal phenotype was resolved by a second ophthalmology-trained retinal specialist (TLE).

### 2.3. Genetic Testing

In Perth, genetic testing for *RPGR* carriers was conducted through the Australian Inherited Retinal Disease Registry and DNA Bank. The methods used included using targeted Sanger sequencing or quantitative polymerase chain reaction (qPCR) to detect familial disease-causing variants in the *RPGR* gene, and they were performed by Molecular Vision Laboratory (Oregon, Hillsboro, OR, USA) and the Australian Genome Research Facility (Perth, Australia) [18]. Additionally, four obligate carriers who did not have genetic testing results but had a family history and clinical signs of XLRP were included in this study.

Genetic testing of carriers in Melbourne was performed using clinical-grade next generation sequencing panels of 314 known IRD genes (Retinal Dystrophy Panel, Blueprint Genetics, Helsinki, Finland). Genetic testing was not performed for healthy controls.

### 2.4. Data Analysis

Visual function was recorded in the form of Early-Treatment Diabetic Retinopathy Study (ETDRS) letters for both best-corrected visual acuity (BCVA) and low-luminance visual acuity (LLVA) (converted into logMAR units). These measurements were then employed to calculate low-luminance deficit (LLD), determined by subtracting LLVA letters from BCVA letters [19]. Retinal function for all carriers was evaluated by obtaining average threshold in decibels (dB) from Macular Integrity Assessment (MAIA) output files and obtaining volumetric measures of hill of vision (HoV) using an online open-source program, https://ocular.shinyapps.io/MAIA3D/ (accessed on 18 February 2024) [20]. Any retinal points with values less than 0 dB were adjusted to −1 dB for analysis following procedures outlined previously [21].

Optical coherence tomography (OCT) images acquired using the Spectralis system (Heidelberg engineering) were exported in XML format, and retinal layer segmentation was conducted using Orion software v3.0 (Voxeleron, LLC, Pleasanton, CA, USA). This analysis enabled calculation of retinal thickness for each B-scan and pixel along the horizontal plane. Subsequently, the thickness data in CSV format generated using Orion were processed using Python (PyCharm CE 2023.2, JetBrains, Prague, Czech Republic) to determine inner-retinal thickness ((IRT) measured from inner limiting membrane to the outer surface of the outer plexiform layer) and photoreceptor complex thickness ((PRC) measured from the inner border of the outer nuclear layer to the inner margin of the retinal pigment epithelium (RPE)) within concentric rings centred on the fovea at 1°, 3°, 5°, and 7° intervals, as previously described [11]. In instances where OCT scans were decentred, resulting in incomplete 7° rings, analysis was limited to the 1–5° rings.

### 2.5. Statistical Analysis

Participant characteristics are reported according to retinal phenotype (healthy control, normal, radial, focal pigmentary retinopathy, and male phenotypes). The distribution of continuous variables is summarised according to the means and standard deviations for variables with an approximately normal distribution and medians and interquartile ranges (IQRs) for non-normally distributed variables. A Mann–Whitney test was used to compare age, spherical equivalent refractive error, and BCVA between healthy controls and *RPGR* carriers.

The 95% limits of agreement were estimated for the inter-rater reliability of retinal grading and agreement between right and left eyes for both BCVA (logMAR) and retinal sensitivity (average dB threshold), and these limits were visually examined using Bland–Altman plots, as previously described [22]. Weighted kappa [23] statistics were used to assess the inter-rater reliability for retinal *RPGR* phenotype classification. To account for the non-independence of data from both eyes, mixed-effects models were applied to compare visual function and retinal structure variables between healthy controls and *RPGR* carriers. Additionally, these models were utilised to compare retinal and visual measurements between carriers with ORF15 and exon 1–14 variants and healthy controls. Age, considered a potential confounder in the classification–outcome relationship a priori, was included as a covariate in all mixed-effects models for all clinical outcome measures, comparing *RPGR* variants with different retinal disease severities with healthy controls and carriers who had ORF15 variants with those with exon 1–14 variants.

Statistical analyses were performed using Prism v10.0 (GraphPad Software, San Diego, CA, USA) and Stata/BE v18 (StataCorp, College Station, TX, USA). An α level of 0.05 was used to indicate statistical significance.

## 3. Results

Seventy eyes of 35 *RPGR* carriers (genetically and/or clinically confirmed) were included in this study (Melbourne: 14 carriers; Perth: 21 carriers). Research visits were conducted between 2013 and 2023. Participant demographics are summarised in Table 1 and detailed in Appendix A. The two independent graders had good agreement for classifying retinal disease severity (weighted kappa = 0.74, 95% confidence intervals: 0.619 to 0.86), and consensus was achieved with the arbitration of the third grader in all cases.

Most carriers (91%) had genetic test results that allowed us to confirm the presence of a pathogenic disease-causing variant according to the American College of Medical Genetics (ACMG) guidelines. The remaining 9% were obligate carriers and therefore diagnosed based on family history (i.e., a family member’s genetic test results) and clinical signs of *RPGR*-associated XLRP. Twenty-six of the thirty-two genotyped carriers (81.25%) had a mutation in the ORF15 region.

### 3.1. Clinical Test Results

*Visual function.* Refractive error, BCVA, LLVA, and LLD were measured for the *RPGR* carriers grouped by retinal phenotype and compared against healthy controls (Figure 2; Appendix A). On average, *RPGR* carriers with focal pigmentary retinopathy had a greater myopic refractive error compared to the healthy controls (approximately 3 dioptres more myopia, *p* < 0.01). BCVA was significantly reduced in *RPGR* carriers with focal pigmentary retinopathy or male pattern phenotypes compared to those with milder phenotypes (*p* < 0.05) and healthy controls (*p* < 0.001).

Furthermore, *RPGR* carriers with focal pigmentary retinopathy or male pattern phenotypes had significantly reduced LLVA compared to those with milder retinal phenotypes (*p* < 0.05) and healthy controls (*p* < 0.001). On average, LLD was not substantially different between *RPGR* carriers and healthy controls (*p* > 0.05), except for *RPGR* carriers with male pattern disease. After adjusting for retinal phenotype, age was not found to be a significant factor affecting BCVA and LLVA (*p* > 0.05), demonstrating that age has no effect on these parameters. Although age was found to be a statistically significant factor affecting refractive error, this was not clinically significant (0.06 dioptres, *p* = 0.007).

*Retinal function.* Fundus-tracked microperimetry was used to investigate retinal function (Figure 3, Appendix A). Retinal sensitivity, as defined by average threshold and HoV volume, was significantly reduced in *RPGR* carriers with focal pigmentary retinopathy or male pattern phenotypes compared to all other milder retinal phenotypes (*p* < 0.05) and healthy controls (*p* < 0.001). Increasing age was found to significantly reduce visual function, as evidenced by declines in both average threshold (−0.05 dB/year, *p* = 0.037) and HoV volume (−11 dB-degrees^2^/year, *p* = 0.032).

*Optical coherence tomography (OCT).* Two eyes from two *RPGR* carriers with a male pattern phenotype had incomplete 7° rings due to decentration of the OCT scan, and therefore only the central 1–5° rings were assessed in the respective eye for these two participants. There were no signs of cystoid macular oedema or epiretinal membranes in any of the carriers. *RPGR* carriers with a male pattern phenotype displayed significant inner-retinal thinning (in comparison to the healthy controls) in the central 3–7° from the fovea, and all carriers with milder retinal disease displayed this feature at 3–5° from the fovea (Figure 4, Appendix A). Notably, retinal remodelling was a common feature for *RPGR* carriers with focal pigmentary retinopathy and male pattern phenotypes demonstrating inner-retinal thickening when comparing carriers with healthy controls. The central 1° appeared particularly susceptible to retinal remodelling, showing inner retinal thickening in both the focal pigmentary retinopathy and male pattern phenotypes, in comparison to the healthy controls, with additional variations observed between the two phenotypes. However, remodelling of the inner retina was less pronounced in the more peripheral retina across all phenotypes. Furthermore, the PRC exhibited thinning with increased retinal disease severity grading and was thinner for all retinal phenotypes at distances ranging from 1 to 7° from the fovea in comparison to the healthy controls. Age did not significantly affect IRT and PRC thickness at any distance from the fovea (*p* > 0.05), with the exception of IRT at 7 degrees from the fovea, which showed a decrease of −0.3 microns/year (*p* = 0.004).

### 3.2. Inter-Eye Agreement for RPGR Carriers and Healthy Controls

The healthy controls exhibited good agreement in BCVA between their right and left eyes (mean difference: −0.01 logMAR units; 95% limits of agreement: −0.14–0.12 (Figure 5A)). *RPGR* carriers had a relatively greater difference between their right and left eyes (mean difference: −0.09 logMAR units; limits of agreement: −1.54–1.71). Carriers with poorer BCVA also tended to have greater differences between the logMAR units of their right and left eyes.

There was good agreement for the MAIA average threshold between the eyes of healthy controls (mean difference: 0.06 dB; 95% limits of agreement: −2.09–2.22 dB (Figure 5B)). *RPGR* carriers had greater differences between their eyes on average (−0.43 dB), which became worse as the average retinal threshold decreased (95% limits of agreement: −6.21–5.35).

There was excellent agreement between right and left eyes for retinal classification (weighted kappa: 0.822, 95% confidence interval: 0.554–0.909).

### 3.3. Genotype–Phenotype Correlations

Genetic variants for *RPGR* carriers were mapped across the *RPGR* gene (Figure 6). Most carriers (n = 26, 81.25% of genotyped *RPGR* carriers) had their genetic variant located on the ORF15 site. The most common (n = 27/32, 83%) type of genetic mutation was a frameshift mutation; these mutations were mostly (n = 26/27, 96%) located in the ORF15 region. In the current cohort of *RPGR* carriers, there were no genotype–phenotype correlations relating to the type or location of mutation on the *RPGR* gene with a retinal disease phenotype. The DNA methylation sites (i.e., CpG sites) for the first 15 exons were also mapped. For all the 31 CpG sites, none of the carriers had genetic variants at any of these sites. Therefore, genetic variants at the DNA methylation sites were not found to influence the phenotypic expression of *RPGR* carriers in the current cohort.

*Influence of variant location for retinal and visual biomarkers.* Clinical and retinal outcome measures were compared between carriers with a genetic variant in the ORF15 site (n = 25 carriers, and 1 carrier was excluded due to a lack of vision tests, multimodal imaging, and microperimetry), those with genetic mutations elsewhere on the gene (i.e., exons 1–14; n = 6 carriers), and the healthy controls (n = 30; Appendix A). *RPGR* carriers with ORF15 variants had reduced LLVA (*p* = 0.018), greater IRT at 1° (*p* = 0.011), and thinner PRC at 7° (*p* = 0.041) compared to those with exon 1–14 variants and the healthy controls (*p* < 0.001). Additionally, *RPGR* carriers with mutations in the ORF15 site exhibited greater myopia (*p* = 0.044) and reduced BCVAs (*p* < 0.001), average retinal thresholds (*p* = 0.006), and HoV volumes compared to the healthy controls (*p* = 0.006). Conversely, *RPGR* carriers with mutations between exons 1 and 14 showed no significant variations from healthy controls except a thinner PRC between 3 and 7° from the fovea (*p* < 0.05). Although there were clinically significant differences in several parameters between the controls and RPGR carriers with mutations on exons 1–14, there was insufficient power to depict these differences statistically. However, these findings potentially suggest that *RPGR* carriers with mutations in the ORF15 site may have worse clinical outcomes, as demonstrated in the male population [24]. Moreover, LLVA, IRT thickness at 1°, and PRC in the midperiphery (i.e., 7°) could serve as sensitive outcome measures for distinguishing between *RPGR* carriers with and without mutations at the ORF15 site.

## 4. Discussion

This multisite cross-sectional study investigated the genotype and phenotype characteristics of an Australian cohort of *RPGR* carriers, aiming to determine appropriate biomarkers for future interventional trials. The prevalence of ORF15 mutations in our cohort aligns with previous reports, ranging from 60 to 73% in both affected males and female carriers [15,25,26].

LLVA, average retinal sensitivity, and HoV volume were significantly reduced in *RPGR* carriers with severe disease compared to those presenting with milder disease. Analysis of IRT and PRC thickness showed general thinning at all distances from the fovea, except for thickening observed at the central 1° from the fovea in *RPGR* carriers with focal pigmentary retinopathy and/or a male pattern phenotype, compared to the healthy controls. A comprehensive evaluation of genotype–phenotype correlations revealed that although there may not be a direct link between the genotype and the severity of the retinal disease phenotype, *RPGR* carriers with disease-causing variants at the ORF15 site appeared to have poorer clinical outcomes, particularly concerning LLVA and retinal thickness, compared to carriers with exon 1–14 variants, irrespective of age. This observation may be linked to the tendency of ORF15 variants, particularly those in the C-terminal basic domain, to be associated with cone-dominated *RPGR* phenotypes that exhibit minimal rod involvement [24]. Consequently, clinical outcomes affecting the central retina, such as retinal thickness and visual acuity, are likely to be more pronounced in carriers with ORF15 variants compared to those with exon 1–14 variants.

The association between myopia and *RPGR* carriers has been well documented in the past; however, conflicting reports exist, with some studies reporting inter-eye symmetry [27,28] and others asymmetry [15,25,29] of myopia. The *RPGR* carriers in the current study exhibited significant increases in myopia in the left eye compared to healthy controls, though this disparity was not observed in the right eye, and in neither case was the level of myopia pathological. While Li et al. proposed that asymmetric myopia and retinal disease in carriers result from X inactivation at the tissue level [28], further genomic investigations are should be carried out to confirm this. Studies exploring genotype correlations concerning myopic presentation have indicated that affected males with disease-causing variants in the ORF15 site exhibit a higher prevalence of myopia [2,30], whereas females with mutations in exons 1–14 have asymmetric myopia and poorer vision [25,28,31]. Given the limited sample size of carriers with genetic variants in exons 1–14, our study found no significant differences in myopic refractive error based on genetic variant location (i.e., ORF15 vs. exon 1–14 variants); however, a higher refractive error was observed for carriers with ORF15 site variants in comparison to the controls. Additionally, BCVA was statistically reduced in the *RPGR* carriers in comparison to the healthy controls, although this reduction was not clinically significant, as the carriers maintained an average BCVA of 20/20, whereas controls had an average of 20/16.

In the current study, *RPGR* carriers exhibited almost perfect agreement in terms of retinal disease severity between eyes, a phenomenon consistent with prior research findings [25,26]. However, there was limited agreement for BCVA between the right and left eyes of *RPGR* carriers, particularly in cases of reduced BCVA. Talib et al. previously reported a strong inter-eye correlation with respect to BCVA among 125 *RPGR* carriers, with only a minority exhibiting inter-eye asymmetry (26%) [25]. Furthermore, inter-eye symmetry with respect to the retinal sensitivity of *RPGR* carriers has not been explored previously; however, affected males have exhibited great symmetry in terms of the average threshold [32,33,34]. Our study demonstrates a marked symmetry in average retinal sensitivity among *RPGR* carriers, similar to that of affected males.

Recent studies indicate that in microperimetry, the retinal sensitivity (assessed via HoV volume) surpasses average threshold measurements for evaluating retinal function in IRDs [20,34]. Unlike the average threshold, HoV volume remains unaffected by data distribution skewing among affected individuals. Our team has similarly observed that HoV volume demonstrates superior sensitivity in discerning various degrees of retinal disease severity in carriers of *CHM* variants in comparison to the average threshold [17]. However, this disparity was not observed in *RPGR* carriers, for whom both the average threshold and HoV volume yielded similar results. One possible explanation for this discrepancy lies in the widespread loss of retinal function across the retina in *RPGR*-associated RP, with central atrophic changes occurring only with a male pattern phenotype [34]. In contrast, *CHM* carriers are typically classified as having intermediate disease characterised by the presence of atrophy, irrespective of its geographic location. These distinctions in disease pathophysiology and atrophy distribution disparity may have contributed to the absence of skewed data in our *RPGR* carrier cohort. Furthermore, in studies or clinical evaluations focusing on *RPGR* carriers with a male pattern phenotype, there may be a preference for utilising HoV volume for diagnosis and monitoring. This choice is motivated by the skewed data distribution and the desire to reduce the averaging effects inherent in average threshold measurements.

LLVA has also emerged as a sensitive indicator of macular visual function in *RPGR*-associated RP and choroideremia [19]. In a previous study, LLVA was found to be able to reveal improvements in visual function following low-dose cotoretigene toliparvovec gene therapy in *RPGR*-associated RP males [35]. In the current study, LLVA was found to diminish with an increasing retinal disease severity in *RPGR* carriers, making it a valuable biomarker for distinguishing between different retinal phenotypes of *RPGR* carriers. However, once atrophic changes manifested (i.e., a male pattern phenotype), BCVA was equally sensitive. Hence, depending on whether a clinical test is employed for diagnostic or therapeutic efficacy, both LLVA and BCVA can complement each other and thus provide a comprehensive evaluation of visual function in *RPGR* carriers. This further highlights the importance of accurately grading the phenotypic classification in carriers. However, more work is required to investigate the impact of treatment on these measures in female carriers.

Retinal thickness, measured via IRT and PRC thickness, has also been thoroughly investigated in RP [10]. Jones et al. delineated three phases of retinal degeneration and remodelling, spanning from photoreceptor stress signalling to rewiring and neuronal death [10]. While extensively studied in affected males, there is lack of research on these changes in female carriers. Following the methodology outlined by Jolly et al., our study investigated retinal remodelling in *RPGR* carriers. In Jolly et al.’s previous paper, inner-retinal thickening and PRC thinning of the central 1–7° (in comparison to controls) were reported in affected males, and these issues worsened upon the onset of severe retinal disease [11]. Conversely, *RPGR* carriers with a male pattern phenotype and/or focal pigmentary retinopathy showed inner retinal thickening in comparison to the controls only within the central 1°. Furthermore, *RPGR* carriers with focal pigmentary retinopathy exhibited the most pronounced thickening compared to those with the male pattern phenotype. As a degenerative condition, generalised thinning of the retina is expected; however, thickening of the inner retina is consistent with retinal modelling, as previously detailed [10]. This finding reported in the current study potentially suggests that retinal remodelling occurs earlier in milder disease stages, while atrophic changes characteristic of male pattern disease lead to inner-retinal degeneration, contrasting with the former phenotype.

Extensive research into genotype–phenotype correlations has been conducted for males with *RPGR*-associated RP and the retinal phenotype in female carriers. Despite reports of correlations [9,28,32], Bellingrath et al. found no correlation between variant location and visual acuity or perimetry in affected males [26]. Similarly, Di Iorio et al. observed no significant differences in disease severity, refractive error, or photopic electroretinography (ERG) between ORF15 and exon 1–14 variants in affected males but noted faster BCVA loss with ORF15 variants (2 ETDRS letters/year vs. 0.5 ETDRS letter/year) [36]. Several studies also report no correlation between BCVA and/or retinal phenotype in *RPGR* carriers with ORF15 and exon 1–14 variants [4,25,37]. The genotype–phenotype assessments in the current study support previous findings that there are no correlations between disease-causing variants and retinal disease severity. However, *RPGR* carriers with variants at the ORF15 site had significant impacts on specific outcome measures such as LLVA, IRT at 1,° and PRC thickness at 7° compared to exon 1–14 variants, constituting an area that has not been previously explored.

### Limitations

The current study utilised FAF imaging for retinal phenotype grading, as recommended by Nanda et al. [4]. Future research should incorporate ERG to pinpoint the specific retinal cells affected, thereby offering more-detailed insights into disease pathogenesis and mechanisms, such as distinguishing between rod–cone and cone–rod dystrophies. Although we included family members sharing the same genotype in the analysis comparing ORF15 and exon 1–14 variants, this factor is unlikely to have influenced biomarker comparisons, as no correlations with retinal phenotype were observed. Moreover, multiple family members with the same genetic variant exhibited varying retinal disease severities. This highlights the potential need to assess X-inactivation skewing ratios in future studies to complement the clinical and genetic data reported for female carriers of X-linked IRDs. Another limitation of the current study is the small number of carriers with the male pattern phenotype (n = nine eyes of five carriers). Nevertheless, this prevalence aligns with the frequency reported by Nanda et al. (n = 13%) in their retinal-phenotype-grading study [4]. Future investigations should thoroughly examine RPGR carriers with the male pattern phenotype and compare and contrast this phenotype with affected males. Additionally, the small number of RPGR carriers with variants in exons 1–14 restricts the ability to generalise the observed associations between clinical outcomes and variant location. However, the carrier cohort in this study is consistent with that in previous reports, where the majority of carriers (66–74%) had genetic variants detected in the ORF15 region [25,38]. Consequently, these findings should be interpreted with caution, and larger-scale studies are needed to validate and expand upon these results. Finally, genetic testing was not conducted on healthy controls to confirm the absence of any IRD-related variants. Given that some women may exhibit subclinical retinal disease, we examined the allele frequencies of the identified variants in the Genome Aggregation Database (gnomAD). While most variants were absent from the database, those that were present had a frequency of 0.0001–0.0004% (Appendix A). Future studies should incorporate genetic testing for control participants to definitively rule out the presence of potentially relevant variants.

Our team has previously emphasised the necessity of additional investigations concerning the interaction between transgene and wildtype genes in female X-linked IRD carriers [17]. Dyka and Molday previously evaluated the protein assembly and co-expression of *RS1* wildtype and missense variants together with myc-tagged wildtype *RS1* genes (i.e., transgene) in an in vitro study to determine the potential effect of gene therapy in carriers of X-linked retinoschisis [39]. However, this did not lead to any in vivo studies or validation in other X-linked conditions, including *RPGR*-associated RP. It is crucial to identify any potential interactions that may lead to abnormalities in protein assembly and function if gene therapy has been initiated for female carriers of X-linked retinal diseases. This work will be important to prove the safety and efficacy of such interventions.

## 5. Conclusions

This cross-sectional study presents the clinical characteristics of a sizable cohort of *RPGR* carriers in Australia. Significant variability exists among different retinal phenotypes, which can be discerned using sensitive biomarkers, such as BCVA, LLVA, retinal sensitivity, and retinal thickness, depending on the extent of central retina involvement. Therefore, such parameters should be considered as a useful outcome measure for determining which participants (i.e., those with moderate to severe retinal disease phenotypes) are eligible for participation in future clinical trials and/or the efficacy of treatment. While no genotype–phenotype correlations were identified between the retinal phenotypes of *RPGR* carriers with genetic variants in the ORF15 site and those with variants in exons 1–14, this study revealed significant changes in LLVA, IRT at 1°, and PRC thickness at 7° in carriers with ORF15 variants. Further investigations will help identify any consequences of introducing the transgene to cells expressing either the wildtype or mutated genes before such therapeutic interventions can become widespread.

## Figures and Tables

**Figure 1 genes-16-00221-f001:**
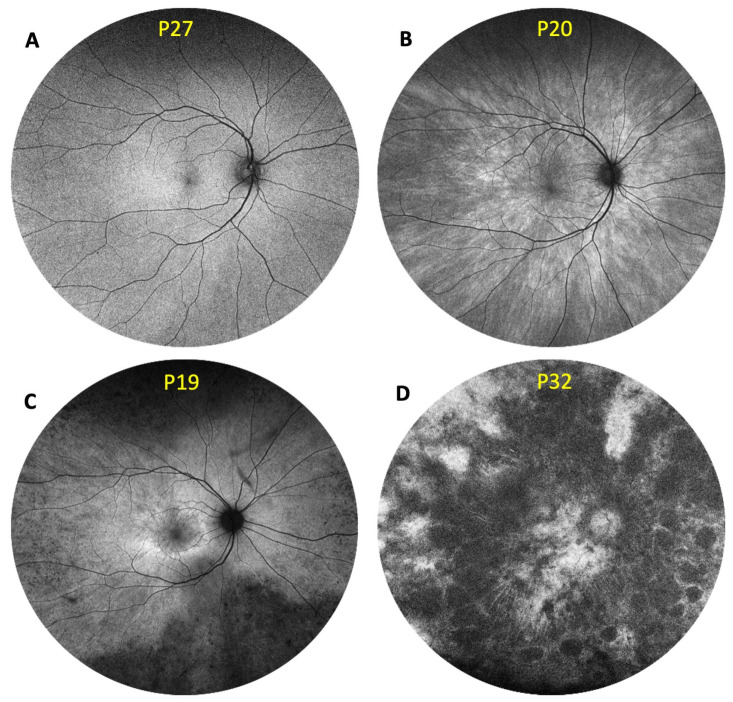
Retinal disease severity of *RPGR* carriers. Retinal disease classification was based on fundus autofluorescence imaging using Optos ultrawide retinal photography, as previously described by Nanda et al. (2018) [4]: (**A**) normal; (**B**) radial; (**C**) focal pigmentary retinopathy; and (**D**) male phenotype.

**Figure 2 genes-16-00221-f002:**
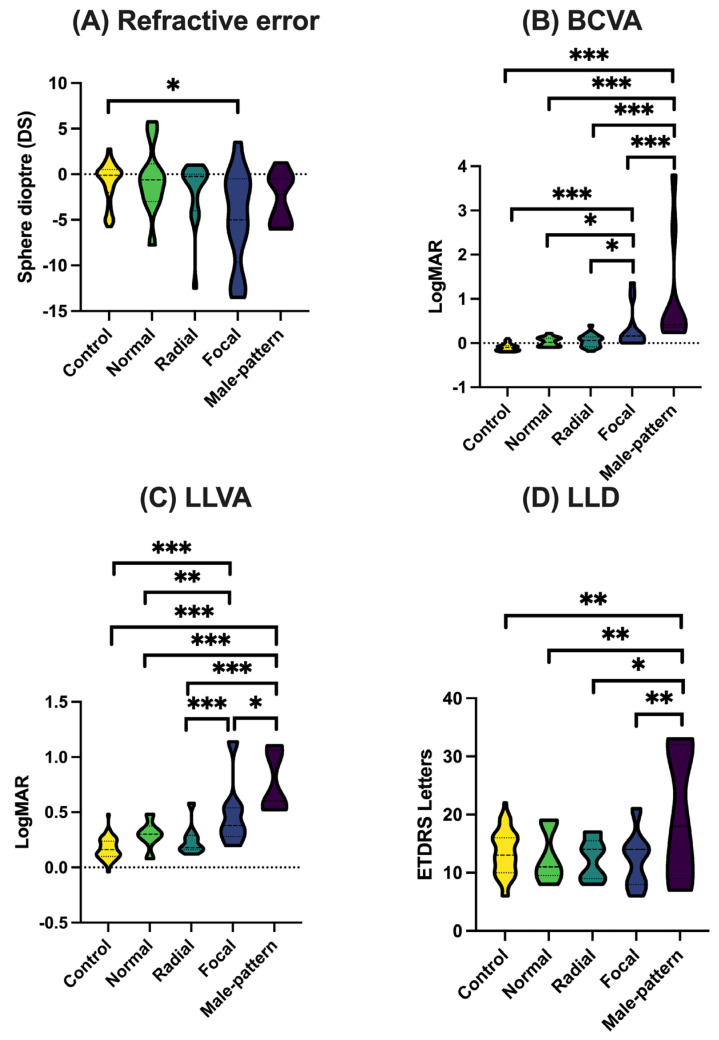
Visual function of *RPGR* carriers compared to that of healthy controls. Tests of visual function involved (**A**) spherical equivalent refractive error, (**B**) best corrected visual acuity (BCVA), (**C**) low luminance visual acuity (LLVA), and (**D**) low luminance deficit (LLD). BCVA and LLVA can distinguish between different disease severities. * *p* < 0.05; ** *p* < 0.01; *** *p* < 0.001 (adjusting for age).

**Figure 3 genes-16-00221-f003:**
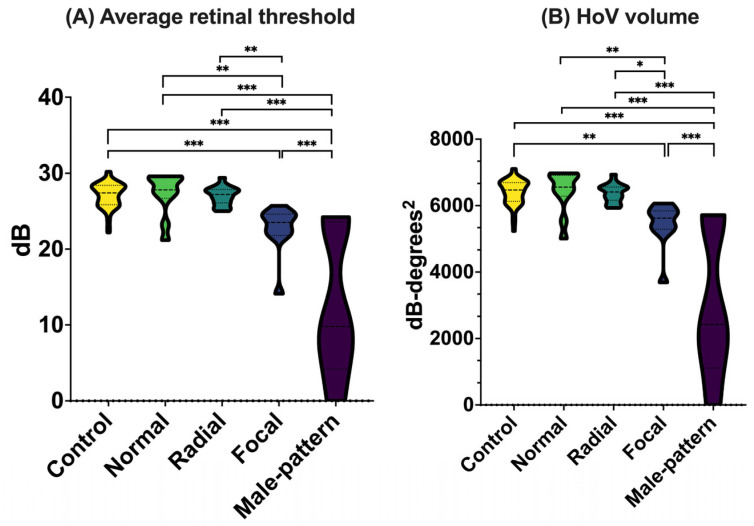
Retinal function of *RPGR* carriers and healthy controls, as assessed via MAIA microperimetry. *RPGR* carriers grouped by retinal disease severity (normal, radial, focal, and male pattern) were compared to healthy controls regarding (**A**) average retinal threshold and (**B**) hill of vision volume. Abbreviations: dB, decibels; HoV, hill of vision. * *p* < 0.05; ** *p* < 0.01; *** *p* < 0.001, age adjusted.

**Figure 4 genes-16-00221-f004:**
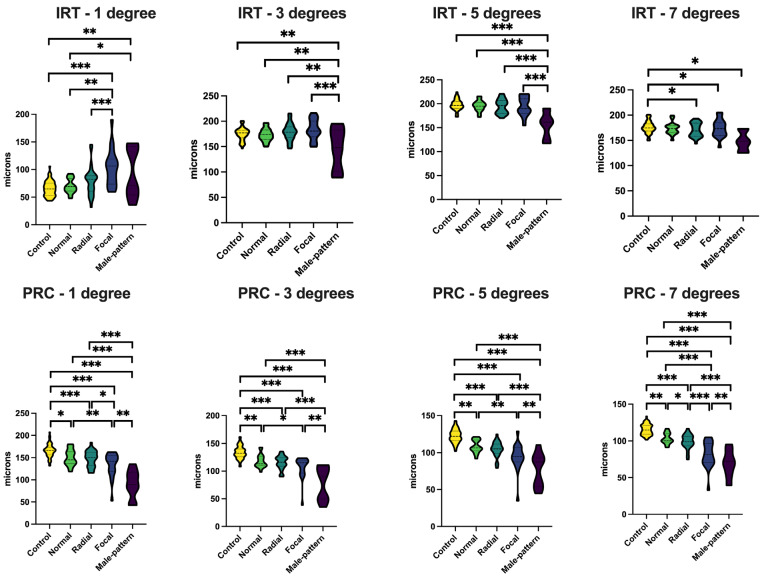
Retinal thickness analysis of *RPGR* carriers and healthy controls for the central 1–7° from the fovea. Retinal thickness was defined as inner retinal thickness (top row) and photoreceptor complex thickness (bottom row). Abbreviations: IRT, inner retinal thickness; PRC, photoreceptor complex. * *p* < 0.05; ** *p* < 0.01; *** *p* < 0.001.

**Figure 5 genes-16-00221-f005:**
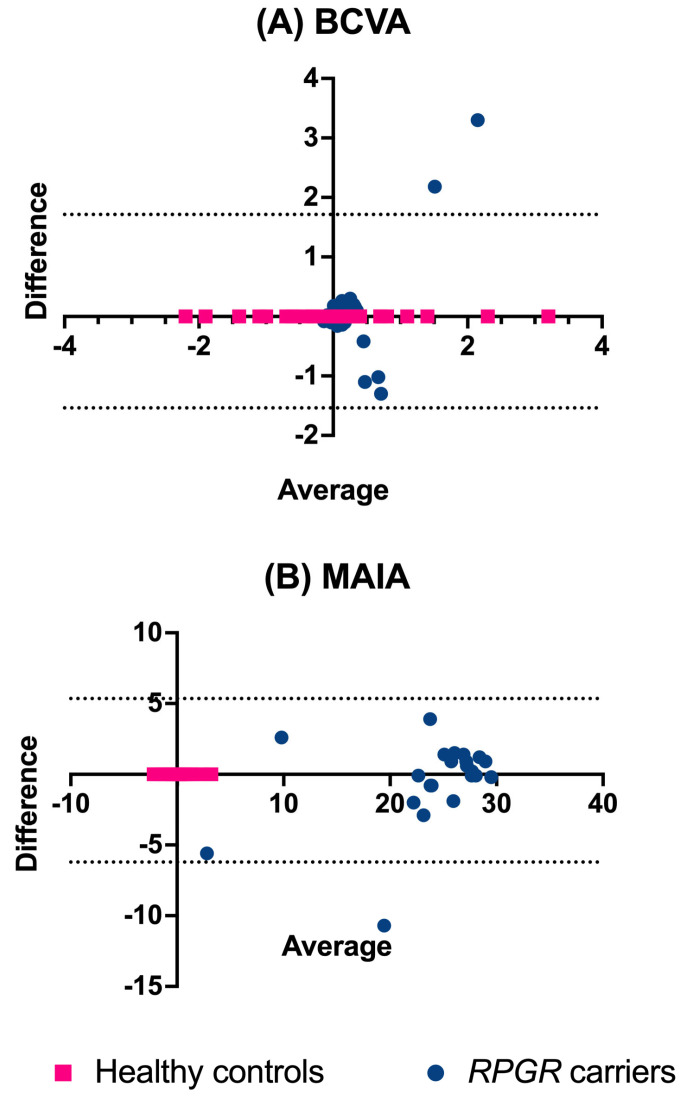
Bland–Altman plot of the agreement between right and left eyes of *RPGR* carriers and healthy controls. Graphs depicted for (**A**) best-corrected visual acuities (BCVAs), logMAR units, and (**B**) retinal sensitivities (dB) for *RPGR* carriers and healthy controls.

**Figure 6 genes-16-00221-f006:**
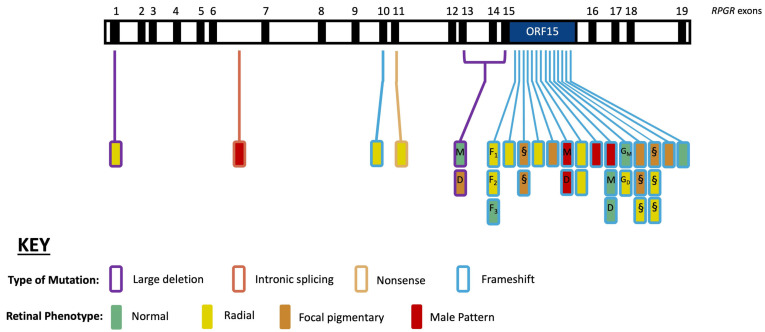
Retinal disease phenotype and genetic variants along the *RPGR^Ex1–19^* gene. Exons 1–19 of the *RPGR* gene are depicted, with each rectangle denoting a female *RPGR* carrier. Vertically stacked rectangles illustrate carriers with the same disease-causing variant, and family members are depicted as follows: D, daughter; G_M_, grandmother; G_D_, granddaughter; M, mother; S, sister. Note: F_1_, F_2_, and F_3_ are members of the same extended family (i.e., F_1_ is the aunt, and F_2_ and F_3_ are first cousins but not daughters of F_1_). The location and colour of the lines indicate the affected exons and type of mutation, respectively. The colour of the rectangle (filled) indicates the retinal disease phenotype (normal, radial, focal pigmentary, and male pattern phenotypes).

**Table 1 genes-16-00221-t001:** Characteristics of the RPGR carriers and healthy controls.

Demographics.	*RPGR* Carriers (n = 35)	Healthy Controls (n = 30)	*p* Value
Age, median (IQR)	40 (26–54) years	48.5 (30–61) years	0.26
Place of residence, n (%)	Melbourne: 14 (40%) Perth: 21 (60%)	Melbourne: 30 (100%)	NA
Retinal phenotype ^a^, n (%)	Normal: 15 (21.43%) Radial: 26 (37.14%) Focal: 20 (28.57%) Male: 9 (12.86%)	No retinal disease: 30 (100%)	NA
Spherical equivalent refractive error, median (IQR)	OD: −0.75 (−4, 0) OS: −1.25 (−5, −0.25)	OD: −0.25 (−2, 0.50) OS: −0.125 (−1.50, 0.50)	0.16 **0.03**
BCVA (logMAR), median (IQR) ^b^	OD: 0.11 (−0.02, 0.26) OS: 0.12 (0, 0.24)	OD: −0.13 (−0.16, −0.06) OS: −0.1 (−0.16, −0.06)	**<0.0001** **<0.0001**
Location of mutation ^c^ ORF15 Exons 1–14 Not tested	26/32 (74.29%) 6/32 (17.14%) 3/35 (11.1%)	NA	

^a^ Retinal disease severity based on classification by Nanda et al. [4]. ^b^ All controls had best-corrected visual acuity (BCVA) tested, and 30 carriers had BCVA recorded (i.e., 6 carriers did not have their vision documented). Thirteen carriers did not have documented refractive error. ^c^ Percentage outlined for total carriers included in the study. Bold values indicate statistically significant *p* values.

## Data Availability

The data supporting this study are not publicly available due to privacy and ethical restrictions.

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
