# Peer review of "Retinal Disease Variability in Female Carriers of RPGR Variants Associated with Retinitis Pigmentosa: Clinical and Genetic Parameters"

_genes, 2025, doi:10.3390/genes16020221_

Round 1

Reviewer 1 Report

Comments and Suggestions for Authors

In this paper, the authors investigate visual function, retinal features, and genotype-phenotype correlations in an Australian cohort of RPGR carriers. The study is comprehensively described, and I particularly appreciate Figure 6, which simplifies the interpretation of the results.

I suggest one additional revision to enhance the genetic evaluation of the data. Healthy controls included in the study were not subjected to genetic analyses (as mentioned on page 4 of 16). To address this limitation and bridge the gap between cases and controls, I recommend conducting a more extensive evaluation of the mutations identified in the patient group. Specifically, the authors should assess the GnomAD frequency of all reported mutations to estimate the prevalence of positive samples in the control group. This analysis is particularly crucial given that the phenotype in carrier women is often subtle or not readily apparent.

Please include this additional evaluation in the relevant sections of the manuscript: Methods, Results, and Discussion.

Author Response

Reviewer 1: In this paper, the authors investigate visual function, retinal features, and genotype-phenotype correlations in an Australian cohort of RPGR carriers. The study is comprehensively described, and I particularly appreciate Figure 6, which simplifies the interpretation of the results. I suggest one additional revision to enhance the genetic evaluation of the data. Healthy controls included in the study were not subjected to genetic analyses (as mentioned on page 4 of 16). To address this limitation and bridge the gap between cases and controls, I recommend conducting a more extensive evaluation of the mutations identified in the patient group. Specifically, the authors should assess the GnomAD frequency of all reported mutations to estimate the prevalence of positive samples in the control group. This analysis is particularly crucial given that the phenotype in carrier women is often subtle or not readily apparent. Please include this additional evaluation in the relevant sections of the manuscript: Methods, Results, and Discussion.

Our response: We thank Reviewer 1 for their comments. As per their suggestion, we have now included the GnomAD allele frequency data in Supplementary Table S3, also containing detailed participant characteristics.

Additionally, we have acknowledged this as a limitation in the discussion section:

            “Finally, genetic testing was not conducted on healthy controls to confirm the absence of any IRD-related variants. Given that some women may exhibit subclinical retinal disease, we examined the allele frequencies of the identified variants in the Genome Aggregation Database (gnomAD). While most variants were absent from the database, those that were present had a frequency of 0.0001-0.0004% (Supplementary Table S3). Future studies should incorporate genetic testing for control participants to definitively rule out the presence of potentially relevant variants.” Page 14

Reviewer 2 Report

Comments and Suggestions for Authors

The authors present a cross-sectional study evaluating retinal alterations in female carriers of RPGR mutation. The manuscript is well structured and the research has been conducted adequately. However, I would suggest some changes:

1) It is not a case-control study, but a cross-sectional. There is no follow-up of patients, so it cannot be a case-control, nor a cohort study.

2) Methods. The type of OCT used should be included in the clinical tests section.

3) All the abbreviations should be explained, such as 'OCT', 'BCVA', or 'LLVA'.

4) Where both eyes included in the analysis or only one eye per patient?

Author Response

Reviewer 2: The authors present a cross-sectional study evaluating retinal alterations in female carriers of RPGR mutation. The manuscript is well structured and the research has been conducted adequately. However, I would suggest some changes:

Comment 1: It is not a case-control study, but cross-sectional. There is no follow-up of patients, so it cannot be a case-control, nor a cohort study.

Our Response: We agree with Reviewer 2’s comments and have changed the study to cross-sectional in the abstract and methods sections.  

Comment 2: The type of OCT used should be included in the clinical tests section.

Our response: We have now added in the type of OCT machine used in the methods section:

            “Optical coherence tomography (OCT) images acquired using the Spectralis system (Heidelberg engineering) were exported in XML format, and retinal layer segmentation was conducted using the Orion software (Voxeleron, LLC, Pleasanton, California, USA).” Page 5

Comment 3: All the abbreviations should be explained, such as 'OCT', 'BCVA', or 'LLVA'.

Our response: All abbreviations in the manuscript, including the main text and abstract, have been reviewed and defined upon first use.

Comment 4: Where both eyes included in the analysis or only one eye per patient?

Our response: We have now clarified that both eye data were used in the statistical analysis:

            “To account for the non-independence of data from both eyes, mixed effects models were applied to compare visual function and retinal structure variables between healthy con-trols and RPGR carriers.” Page 5

Round 2

Reviewer 1 Report

Comments and Suggestions for Authors

In this paper, the authors investigate visual function, retinal features, and genotype-phenotype correlations in an Australian cohort of RPGR carriers. The study is comprehensively described, and I particularly appreciate Figure 6, which simplifies the interpretation of the results. 

Authors have adequately addressed suggestions, the manuscript is now feasible for publication. 

Reviewer 2 Report

Comments and Suggestions for Authors

All suggested changes have been performed.